# Steatosis Quantification on Ultrasound Images by a Deep Learning Algorithm on Patients Undergoing Weight Changes

**DOI:** 10.3390/diagnostics13203225

**Published:** 2023-10-17

**Authors:** Adam P. Harrison, Bowen Li, Tse-Hwa Hsu, Cheng-Jen Chen, Wan-Ting Yu, Jennifer Tai, Le Lu, Dar-In Tai

**Affiliations:** 1Research Division, Riverain Technologies, Miamisburg, OH 45342, USA; adam.p.harrison@gmail.com; 2Department of Computer Science, Johns Hopkins University, Baltimore, MD 20818, USA; lbwdruid@gmail.com; 3Department of Gastroenterology and Hepatology, Chang Gung Memorial Hospital, Linkou Medical Center, Taoyuan 33305, Taiwan; echohsuth45@gmail.com (T.-H.H.); k85731@cgmh.org.tw (C.-J.C.); ting800130@gmail.com (W.-T.Y.); jennifertai112@gmail.com (J.T.); 4DAMO Academy, Alibaba Group, New York, NY 94085, USA; tiger.lelu@gmail.com

**Keywords:** quantitative ultrasound, liver steatosis, weight changes, artificial intelligent, deep learning

## Abstract

Introduction: A deep learning algorithm to quantify steatosis from ultrasound images may change a subjective diagnosis to objective quantification. We evaluate this algorithm in patients with weight changes. Materials and Methods: Patients (N = 101) who experienced weight changes ≥ 5% were selected for the study, using serial ultrasound studies retrospectively collected from 2013 to 2021. After applying our exclusion criteria, 74 patients from 239 studies were included. We classified images into four scanning views and applied the algorithm. Mean values from 3–5 images in each group were used for the results and correlated against weight changes. Results: Images from the left lobe (G1) in 45 patients, right intercostal view (G2) in 67 patients, and subcostal view (G4) in 46 patients were collected. In a head-to-head comparison, G1 versus G2 or G2 versus G4 views showed identical steatosis scores (R^2^ > 0.86, *p* < 0.001). The body weight and steatosis scores were significantly correlated (R^2^ = 0.62, *p* < 0.001). Significant differences in steatosis scores between the highest and lowest body weight timepoints were found (*p* < 0.001). Men showed a higher liver steatosis/BMI ratio than women (*p* = 0.026). Conclusions: The best scanning conditions are 3–5 images from the right intercostal view. The algorithm objectively quantified liver steatosis, which correlated with body weight changes and gender.

## 1. Introduction

After successful global hepatitis B vaccination programs and direct anti-hepatitis C viral regimens, both hepatitis B and hepatitis C diseases declined significantly. Non-alcoholic fatty liver disease (NAFLD) is becoming the leading cause of diffuse liver disease [1]. The gold standard for diagnosing liver steatosis remains liver histology studies. Magnetic resonance imaging proton density fat fraction (MRI-PDFF) is also accepted as a non-invasive gold standard alternative [2,3]. However, MRI is quite expensive. The 2D ultrasound (US) is widely used as the first-line diagnostic modality for liver diseases. It may also detect liver steatosis. Unfortunately, the US diagnosis of liver steatosis or fibrosis is quite subjective. Several non-invasive quantitative ultrasounds (QUS) have been developed with increasing popularity [4]. These include control attenuation parameters [5], attenuation imaging [6], ultrasound-guided attenuation parameters [7], tissue attenuation imaging (TAI), tissue scatter-distribution imaging (TSI) [8], and sound speed estimation [9]. Even so, these modalities need specialized equipment that is not always available. Each modality has its own diagnostic condition and is difficult to exchange or compare. Another direction is to apply machine learning on standard 2D B mode US to quantify liver steatosis [10,11,12,13,14,15,16]. Using deep learning (DL) on a big data US cohort, we developed an algorithm that can quantify liver steatosis from US images [10]. This modality has advantages over other QUS in that it does not need an area of interest, can be applied to US images acquired from standard scanners, and can learn from images collected in either retrospective or prospective studies.

A limitation of the previous study was that it was evaluated only on patients with gold-standard biopsies [10], introducing a selection bias in the test cohort. Gaining reliable gold standards from a broader population is difficult due to the invasiveness or expense of the measures. However, it is established that liver steatosis is positively correlated with body weight [17] and can act as a “silver standard”. Therefore, we evaluate our algorithm on a longitudinal cohort of patients who underwent body weight changes to measure how well our algorithm correlates with these changes over time.

## 2. Materials and Methods

This study was approved by the Institutional Review Board (IRB) of the Chang Gung Medical Foundation (CGMH IRB No. 201801283B0 and 202200758B0). The informed consent was waived by IRB because this is a retrospective study.

### 2.1. Patients

Our retrospective cohort included patients undergoing long-term surveillance for chronic liver disease at the Chang Gun Memorial Hospital outpatient department. Among them, patients with a history of weight changes greater or equal to 5% between June 2018 and December 2020 were enrolled in this study. A 5% weight change will decrease hepatic steatosis significantly [18]. The corresponding sequential 2D ultrasound studies between 2013 to 2021 were collected. Two hepatologists (D Tai and T Hsu) with over 20 years of experience assessed images and excluded those of poor quality. Those studies without body weight records or those recorded as dual images were also excluded. Patients with only one single eligible study were also excluded. During the exclusion process, some patients may have weight changes of less than 5% at the timepoints with available ultrasound studies. Therefore, the whole series will be categorized into weight change greater than or equal to 5%, or lower than 5% groups.

### 2.2. Image Views

Eligible images were classified into 4 view groups according to the scanning view as detailed in our previous work [10], except with a slight modification. Briefly, view group 1 (G1) includes images of the left hepatic lobe scanned with either vertical or horizontal views; view group 2 (G2) includes images of the right hepatic lobe scanned at the intercostal space; view group 3 (G3) includes images focused on liver/kidney contrast scanned with either intercostal or subcostal views; and view group 4 (G4) includes images with subcostal view. The only difference with the previous study is that subcostal scans with liver/kidney contrast are not included in G4.

### 2.3. Preparation and Reading of Images

Selected US images were first preprocessed to remove any image regions outside of the actual scan area, excluding personal identification, brand of machine, and study information [10]. After converting the images into PNG files, images were read by our established steatosis deep learning algorithm [10], which we summarize below. Our algorithm was trained on the largest and most diverse cohort of ultrasound images to date for steatosis quantification, totaling 2899 patients and 200,654 images acquired across 13 different scanners. Images were retrospectively mined from the picture archiving and communication system of the Chang Gung Memorial Hospital and came accompanied by steatosis assessments produced from ultrasound readings during routine clinical care. The selection criterion was those patients who received elastography between 2011 and 2018. As backbone, our algorithm used ResNet18 [19] and was trained using an ordinal loss [20]. Once trained, the algorithm can provide a continuous steatosis severity score for an image ranging from 0 to 1, otherwise known as its image-wise score. The algorithm was validated (N = 147) and tested (N = 112) on histology-proven cases, demonstrating area under the curves of the receiver operating characteristic curve to classify mild, moderate, and severe steatosis grades of 0.85, 0.91, and 0.93, respectively. More details can be found in Li et al. [10], including additional analyses on the algorithm’s repeatability across scanner types.

### 2.4. Ultrasound Steatosis Score

In our previous study [10], special care was taken to examine how many images in each view group could produce stable and reproducible data. We found that 3–5 images may provide a max repeatability coefficient lower than 0.3, except for group 3 (Table 1). Therefore, the view group scores were defined as the mean of 3–5 image-wise scores for each view group. Groups with less than 3 images in each study were excluded. Each study should have at least one group with ≥3 eligible images.

Although our prior work derived a steatosis score for each view group [10], for this analysis we must give every study a *single* diagnostic result. Thus, we defined a protocol as follows. Because steatosis scores from the G2 view were the most accurate [10], we selected the G2 view preferentially. We fall back to G1 or G4 views if the G2 view data is not available. To choose between G1 or G4 views, we select those with the largest view of the liver parenchyma and the best image quality. This protocol can be followed prospectively. For our retrospective cohort, we applied this protocol blind to the DL steatosis score, weight changes, or any other clinical information. We exclude G3 images from our protocol, as these are not collected in sufficient quantity in our practice.

### 2.5. Statistical Analysis

Patient characteristics were represented as the number and percentage, or the mean ± standard deviation (SD) or standard error of mean (SEM), as appropriate. When measuring correlation for serial studies, with repeated measures for each patient, standard correlation measures are not appropriate since they assume independence of error between each observation [21]. Instead, we calculated the repeated measures correlation between patient weights and steatosis scores with the Pingouin Python library [22]. Differences between high and low body weight stages were performed by paired t-tests. Categorical variables were tested using the Chi-square test. Except for the repeated measures correlation, statistical analyses were performed using the SPSS software (version 22; SPSS Inc., Chicago, IL, USA), and a *p*-value of <0.05 was judged as statistically significant.

## 3. Results

### 3.1. View Groups of the Study

Among 101 patients enrolled, 74 patients had adequate images (a total of 2393 images in 239 studies), body weight information, and at least two US studies (mean 3.2, range 2–7 studies/cases). The view groups included in this analysis are shown in Figure 1. G2 had the most eligible cases (67 cases, 184 studies), followed by G1 (45 cases, 116 studies) and G4 (46 cases, 115 studies). G3 had the lowest number of eligible cases (19 cases, 45 studies). The demographics of the study patients are shown in Table 2. Note, even though all patients must have ≥5% weight change in the study period, the removal of studies without enough adequate images can also remove high or low recorded weight timepoints. Thus, after this purging, patients can now have <5% (n = 13) and ≥5% (n = 61) weight changes associated with included imaging studies and corresponding timepoints.

### 3.2. Head-to-Head Comparison of Steatosis Scores between Two Groups in the Same Patient

To understand the difference between groups of the same patient, we collected patients that had both G1 and G2 or G2 and G4 images available. Paired *t*-test studies showed no significant differences in steatosis scores (Table 3). Correlation studies show a high correlation between steatosis scores of the same patient across groups (Figure 2. G1 vs. G2: R^2^ = 0.86, *p* < 0.001; G2 vs. G4: R^2^ = 0.88, *p* < 0.001).

### 3.3. Correlation between Body Weight and Steatosis Score

This cohort included sequential studies. An example of sequential G2 US images of a patient is shown in Figure 3. Steatosis scores from 2D ultrasound studies with different scanners at different timepoints are shown in this figure.

For the whole series, using our study-wise protocol detailed above, 67 patients had sufficient G2 views. For the remainder, we used their G1 or G4 views, as per our protocol. The repeated measures correlation between body weight and steatosis score in all 74 patients is shown in Figure 4. The correlation is quite good (R^2^ = 0.62, *p* < 0.001; 0.50–0.72 95% confidence intervals).

For further analysis, we selected the highest or lowest body weight timepoints among the sequential studies on the same patients. By using the single result of each study, we compared steatosis scores between high body weight and low body weight timepoints (Figure 5). We found that both patients with <5% (*p* < 0.001) and ≥5% (*p* = 0.014) weight changes showed significant differences in steatosis scores between body weight timepoints.

### 3.4. Differences in Steatosis Scores across Gender Differences

Gender differences in liver steatosis in high or low body weight stages are shown in Table 4. In the high body weight timepoints, mean steatosis scores were higher in males, but with no statistical significance. However, the body mass index (BMI) was higher in females. After dividing the steatosis score by BMI, the difference between males and females is statistically significant (*p* = 0.026). Similar situations were present in the low body weight timepoints, but with no statistical significance.

### 3.5. Brands of Ultrasound Scanners Used in Different Viewpoints

Five brands of scanners were used in this study (Table 5). The main scanners were Toshiba TUS-A300 (62%) and Philips iU22 (32.4%).

## 4. Discussion

The steatosis scores quantified from retrospectively collected 2D ultrasound images showed a significant correlation with body weight changes (Figure 4 and Figure 5). These findings provide further important validation that our DL algorithm can be used as an objective quantification of steatosis from 2D ultrasound images [10]. Importantly, the current validation does not have the selection bias of populations with biopsy or MRI-PDFF measurements.

Weight loss induced by lifestyle changes may reduce nonalcoholic steatohepatitis [23]. Covarrubias et al. report an interesting long-term liver and pancreas fat quantification study during a weight-loss surgery program [24]. Nine histology-proven liver steatosis patients were followed up prospectively in a 16-month period after bariatric surgery. They received an MRI-PDFF study during each visit. In the four to five successful MRI-PDFF studies, both liver and pancreas fat fractions decreased in accordance with weight loss. Our retrospective US study included 74 cases. Each case received two to seven qualified 2D US studies. These US studies were not quantitative US. However, our previously established deep learning algorithm, learning from a big data cohort of 2D US images and validated and tested in histology-proven cases [10], made the objective quantitative analysis of this study possible.

Many quantitative US studies using artificial intelligence algorithms developed by 2D ultrasound have been reported [16]. Most of the studies use a single US machine. This strategy may produce lower variations but also severely limit its clinical use. As discussed in our prior work, our algorithm was trained on 13 different scanner models and reports percentage agreement numbers between scanners of 92% or higher. Additionally, most US artificial intelligence studies use a single viewpoint, with a handful of exceptions that use multiple views [14,25,26]. Our algorithm can handle multiple views. These study design choices gave us the opportunity to conduct further retrospective studies, such as this one, that incorporate different scanners (Figure 3) and viewpoints, which allows for a much more flexible and representative data collection strategy.

Different from CT or MRI images, most US operators acquired images without a standardized format. To minimize potential variations, we systematically classified 2D ultrasound images into 4 groups according to the scanning positions [10]. We found that an adequate number of 3–5 images of the right intercostal space can be obtained from this retrospective study (Figure 1, G2, 67 cases, 184 studies). Similar images were taken from G1 (45 cases, 116 studies) and G4 (46 cases 115 studies). Our previous study indicated that G2 images produce the highest repeatability and accuracy [10], while the performances of the G1 and G4 groups are nearly as good (Table 3 and Figure 2). The liver/kidney contrast views (G3) are frequently used for the diagnosis of fatty liver in routine ultrasound scanning. However, only 19 patients and 45 studies had adequate images collected. This view includes two organs that produce higher variations than other view groups [10]. We may need more than five images to achieve acceptable repeatability in this G3 view group.

Based on our previous study [10], we used the mean of three to five images in the same group for maintaining repeatability (Table 1) [10]. Current QUS requires an area of interest [5,6,7,8,9] to maintain the quality of quantification. This procedure is time-consuming and loses other information. For example, the blurring of vessels is not taken into consideration in QUS. In contrast, our DL algorithm accepts whole US images, and it is relatively easy to acquire five images in a short time without interrupting the routine screening procedure. The key points are to acquire good-quality images and to include as much liver parenchyma as possible. For future protocols, we will recommend acquiring five good-quality images of the liver from the right intercostal view for every study. When the right hepatic lobe is not available, five good-quality images from either the left hepatic lobe or subcostal scan will be a good substitution. This standard procedure will create a chance for future retrospective quantification in most patients.

The study was originally aimed at the correlation between steatosis scores with weight changes. Unexpectedly, we found differences in steatosis scores between genders. Women accumulate fat mainly in the subcutaneous adipose tissue while men tend to accumulate fat in abdominal visceral adipose tissue [27,28]. The gonadal steroids, including androgens, estrogens, and progestogens are involved in the control of body fat distribution in humans [29]. A review by Lonardo et al. concluded that the prevalence and severity of NAFLD are higher in men than in women during the reproductive age [30]. This trend decreased after menopause [31]. In this study, we found that men tend to have a higher liver steatosis score than women. Gender steatosis score differences were more prominent in the high body weight than in the low body weight timepoints (Table 4). Identifying the fact that men are more susceptible to liver steatosis than women also supports this DL algorithm as a good quantification model.

It should be noted that weight loss is not the only factor that determines liver steatosis. Oh et al. collected patients with NAFLD who received different therapies [32]. Among them, 45 patients received an exercise regimen and 29 patients received a weight reduction regimen. In the exercise group, their total energy intake was increased with little weight loss, yet they produced a significant reduction in liver steatosis [32]. In Figure 3 of this study, we can see the steatosis score is not completely correlated with weight. The lowest steatosis score occurred at the beginning of a significant weight change rather than at the lowest body weight timepoint. We do not know whether this was related to exercise or other factors.

Our limitation is that weight was used as a quantification standard. This is a retrospective study; we did not have histology or MRI-PDFF study for validation. The other limitation is that different scanners were used in this study (Table 5). Nonetheless, these scanners were used during the DL algorithm development. As the reported agreement among the three main scanners was more than 92% [10], the effect of the different scanners should be minimal. However, measuring the algorithm’s performance on alternative scanners requires further prospective external validation. Additionally, our case number is relatively small. Further confirmation by a larger series may be needed.

We conclude that our DL steatosis algorithm could be considered an objective quantitative model for liver steatosis. This algorithm can be used in retrospective studies on images with different brands of ultrasound machines.

## Figures and Tables

**Figure 1 diagnostics-13-03225-f001:**
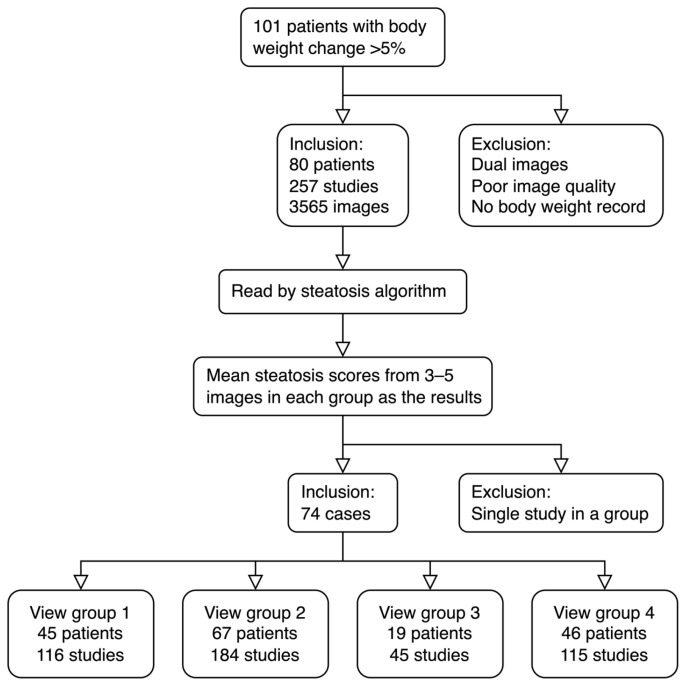
Flowchart.

**Figure 2 diagnostics-13-03225-f002:**
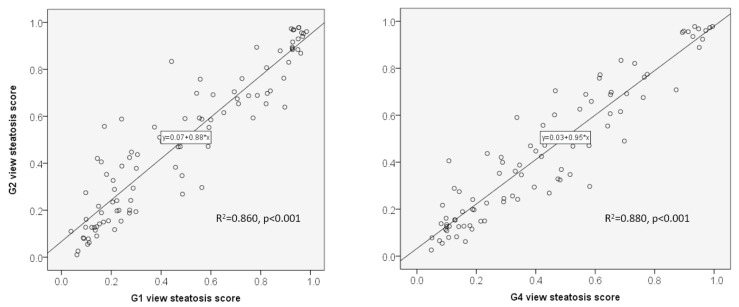
Correlation of steatosis scores between US view groups on the same patient. Left G1 vs. G2 view (R^2^ = 0.86, *p* < 0.001); Right G1 vs. G4 view (R^2^ = 0.88, *p* < 0.001).

**Figure 3 diagnostics-13-03225-f003:**
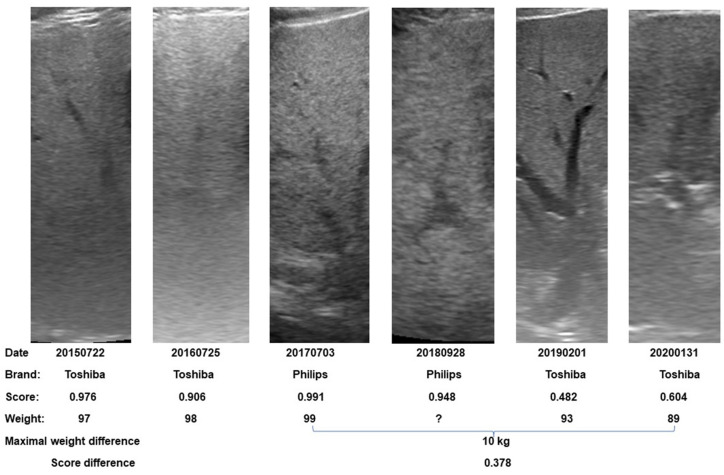
Demonstration of body weight and steatosis score changes by sequential G2 images in a patient. Different scanners were used at different timepoints. A significant decrease in steatosis scores between highest and lowest weight points can be found. One should note that the lowest steatosis score occurred at the beginning of weight change from 99 kg to 93 kg.

**Figure 4 diagnostics-13-03225-f004:**
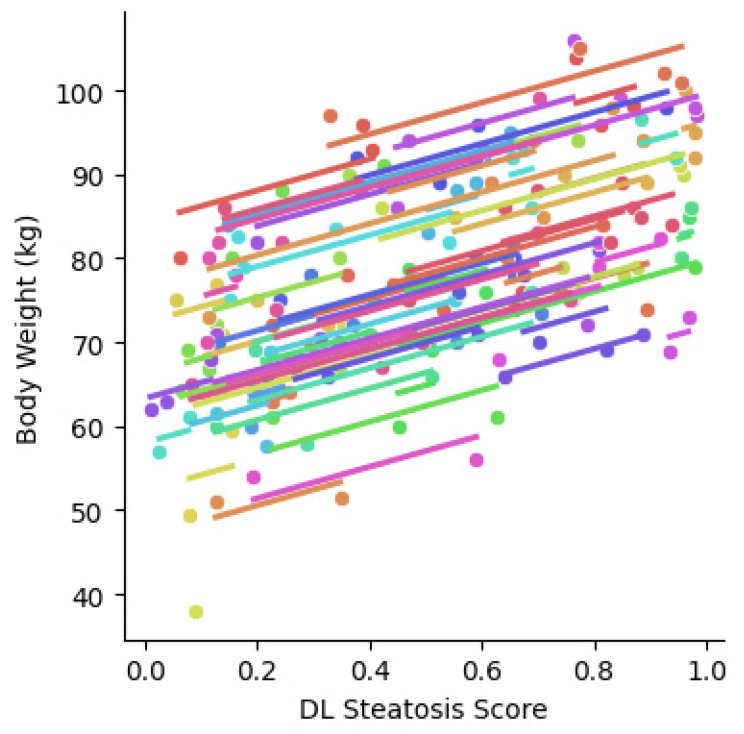
Repeated measures correlation of body weight and DL steatosis scores in the whole series. The analysis was based on data from G2 images. When G2 images are not available, G1 or G4 data with best quality will be added to the analysis. A positive correlation trend is found between weight and steatosis score (N = 74, R^2^ = 0.62; 0.50–0.72 95% CI intervals, *p* < 0.001). Each color represents a different patient.

**Figure 5 diagnostics-13-03225-f005:**
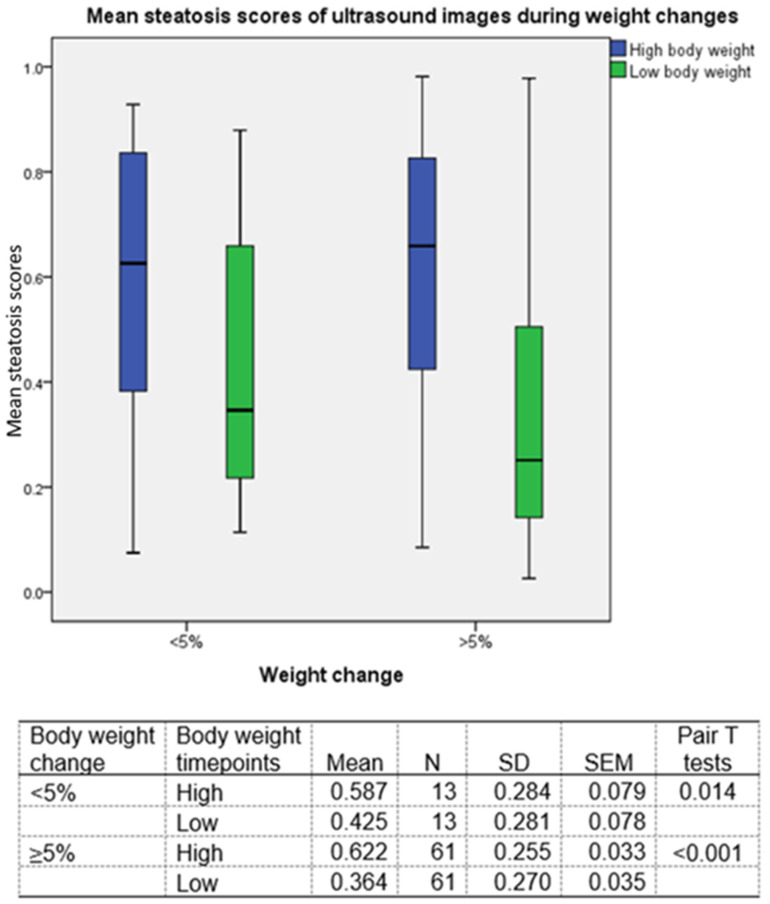
Mean steatosis scores during body weight changes. The highest and lowest body weight stages were determined for each patient. Sixty-one patients had maximal weight differences greater than 5% body weight and thirteen patients had weight differences lower than 5%. A significant mean DL steatosis score difference between highest (*p* < 0.001) and lowest weight timepoints (0.014) can be found in both groups.

**Table 1 diagnostics-13-03225-t001:** The max repeatability coefficient in different numbers of images is tabulated across different view groups.

Group	1 Image	2 Images	3 Images	4 Images	5 Images
1	0.4614 (0.4210, 0.5078)	0.3263 (0.2957, 0.3552)	0.2664 (0.2397, 0.2934)	0.2307 (0.2089, 0.2555)	0.2064 (0.1874, 0.2259)
2	0.3665 (0.3385, 0.3942)	0.2592 (0.2381, 0.2797)	0.2116 (0.1955, 0.2298)	0.1833 (0.1697, 0.1996)	0.1639 (0.1508, 0.1763)
3	0.5264 (0.4711, 0.5814)	0.3722 (0.3292, 0.4135)	0.3039 (0.2717, 0.3393)	0.2632 (0.2350, 0.2906)	0.2354 (0.2117, 0.2596)
4	0.4583 (0.4150, 0.5029)	0.3240 (0.2945, 0.3587)	0.2646 (0.2361, 0.2920)	0.2291 (0.2067, 0.2547)	0.2049 (0.1833, 0.2267)

Parentheses enclose bootstrapped 95% confidence intervals.

**Table 2 diagnostics-13-03225-t002:** Demography.

Category	Number (%) or Mean ± SD
Total No.	74
Gender	
Male	48 (64.9%)
Female	26 (35.1%)
Etiology	
NBNC	26 (35.1%)
HBV	43 (58.1%)
HCV	4 (5.4%)
Alcoholic	1 (1.4%)
Age (year)	51.74 ± 9.85
Initial body height (cm)	165.88 ± 7.88
Initial body weight (kg)	80.74 ± 12.12
Initial AST (U/L)	33.34 ± 18.58
Initial ALT (U/L)	39.90 ± 31.77
Initial total cholesterol (mg)	204.72 ± 58.49
Initial triglyceride (mg)	202.56 ± 141.51

**Table 3 diagnostics-13-03225-t003:** Head-to-head comparison of steatosis scores between view groups.

Image View	Study No.	Steatosis Score			
		Mean	SD	SEM	*p* Value
G1	99	0.481	0.312	0.031	0.377
G2	99	0.491	0.297	0.030	
G2	89	0.441	0.289	0.031	0.357
G4	89	0.431	0.286	0.030	

**Table 4 diagnostics-13-03225-t004:** Gender differences on liver steatosis in studies of G2 view groups.

Body Weight Timepoints	Category	Gender	No	Mean	SD	SEM	*p* Value
High	Steatosis score	M	48	0.6458	0.2573	0.0371	0.173
		F	26	0.5598	0.2560	0.0502	
	BMI	M	48	28.6137	3.3424	0.4824	0.071
		F	26	30.7566	5.3240	1.0441	
	Steatosis score/BMI	M	48	0.0223	0.0084	0.0012	0.026
		F	26	0.0179	0.0073	0.0014	
Low	Steatosis score	M	48	0.3985	0.2847	0.0411	0.307
		F	26	0.3307	0.2417	0.0474	
	BMI	M	48	26.0044	3.1512	0.4548	0.267
		F	26	27.0644	4.9994	0.9805	
	Steatosis score/BMI	M	48	0.0149	0.0100	0.0014	0.112
		F	26	0.0116	0.0074	0.0014	

BMI: body mass index.

**Table 5 diagnostics-13-03225-t005:** Brands of ultrasound scanners used from different viewpoints.

View Groups	Brands					
	Aloka SSD 5500	Hitachi Preirus	Philips iU22	Siemens S2000	Toshiba TUS-A300	Total
1	0	2	32	1	81	116
2	5	6	59	3	110	184
3	0	0	13	1	31	45
4	0	5	45	3	63	115
Total (%)	5 (1.1)	13 (2.8)	149 (32.4)	8 (1.7)	285 (62.0)	(100) 460

## Data Availability

The data are not publicly available due to patient privacy concerns.

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
