# Peer review of "Steatosis Quantification on Ultrasound Images by a Deep Learning Algorithm on Patients Undergoing Weight Changes"

_diagnostics, 2023, doi:10.3390/diagnostics13203225_

Round 1

Reviewer 1 Report

This study appears to have evaluated the degree of hepatic steatosis in patients with follow-up data and changes in body weight by applying an existing algorithm. However, the paper lacks clarity and organization regarding the explanation of the algorithm, the process of its implementation in the study, the specific results produced by the algorithm, and a clear description of the patient population. In particular, it is challenging to distinguish between the methodology used in this study and the processes described in previous research. Additionally, the small sample size of the patient group for testing makes it difficult to assess the validity of the research findings presented by the authors.

Author Response

Dear editors and reviewers,

Thank you very much for your constructive comments and the chance to improve our manuscript.

We revised our submission to address reviewer comments and the point-to-point responses are listed below:

Reviewer 1

This study appears to have evaluated the degree of hepatic steatosis in patients with follow-up data and changes in body weight by applying an existing algorithm. However,

  1. the paper lacks clarity and organization regarding the explanation of the algorithm, the process of its implementation in the study, the specific results produced by the algorithm, and a clear description of the patient population.

We agree that more details on our prior work detailing our algorithm would help strengthen our submission. For this reason, we have added additional details on the patient population, algorithmic development, and results of our prior work. This has given readers better context to interpret our follow-up analysis and has improved the clarity and organization of our presentation. Readers are encouraged to consult our prior work for any additional details.

In addition, we note that we added more information on patient recruitment in the patient section. These patients were enrolled when a history of weight change greater than 5% were found by D Tai.

  1. In particular, it is challenging to distinguish between the methodology used in this study and the processes described in previous research.

Our contribution for this work is our supplemental analysis using weight-change data that is not affected by the selection biases seen in histology-proven data. This is an important analysis rarely seen in other works. Combined with our previous evaluation on histology proven data, this work helps provide a complete picture of our algorithm’s ability to quantitatively assess steatosis.  

The algorithmic methodology remains the same, which we have now mentioned explicitly in our revision. This has clarified our contributions.

  1. Additionally, the small sample size of the patient group for testing makes it difficult to assess the validity of the research findings presented by the authors.

We agree that more case numbers would strengthen our analysis, and we now explicitly highlight this limitation in our discussion. Even so, while we agree that greater sample sizes are always desirable, we do emphasize that our work’s sample size of 239 studies is not worse than other works that focus on automatic steatosis assessment, such as the 55 studies in [1], 63 studies in [2], 135 studies in [3], 204 studies in [4], and 240 studies in [5] (Reference list below). Our correlation trends were statistically significant.

References:

  1. Byra M, Styczynski G, Szmigielski C, et al. Transfer learning with deep convolutional neural network for liver steatosis assessment in ultrasound images. Int J Comput Assist Radiol Surg. 2018;13(12):1895-1903. doi:10.1007/s11548-018-1843-2
  2. Biswas M, Kuppili V, Edla DR, et al. Symtosis: A liver ultrasound tissue characterization and risk stratification in optimized deep learning paradigm. Comput Methods Programs Biomed. 2018;155:165-177. doi:10.1016/j.cmpb.2017.12.016
  3. Byra M, Han A, Boehringer AS, et al. Liver Fat Assessment in Multiview Sonography Using Transfer Learning With Convolutional Neural Networks. Journal of Ultrasound in Medicine. 2021. doi:10.1002/jum.15693
  4. Han A, Byra M, Heba E, et al. Noninvasive Diagnosis of Nonalcoholic Fatty Liver Disease and Quantification of Liver Fat with Radiofrequency Ultrasound Data Using                    One-dimensional Convolutional Neural Networks. Radiology. 2020;295(2):342-350. doi:10.1148/radiol.2020191160
  5. Cao W, An X, Cong L, Lyu C, Zhou Q, Guo R. Application of Deep Learning in Quantitative Analysis of 2-Dimensional Ultrasound Imaging of Nonalcoholic Fatty Liver Disease. Journal of Ultrasound in Medicine. 2020;39(1):51-59. doi:https://doi.org/10.1002/jum.15070

Reviewer 2 Report

In the manuscript, authors used a deep learning algorithm on patients undergoing weight changes to perform the steatosis quantification. It is an interesting and significative study. Some individual comments were provided below:

Major

1)    Why authors collected patients whose weight changes greater than or equal to 5%? please provide some explanation. Similarly, please provide the explanation about the definition of “poor quality” in the sentence “We excluded images with poor quality, without body weight records, or those recorded as dual images.”

2)    Limited sample size is the major limitation of the study. Authors need to provide some explanation in the section of discussion.

Minor

1)Delete the words “(word count 200)” in the end of the abstract.

2)In the methods, authors need to provide more information about the ethics including if the informed consents were obtained from all the participants.

3)Table 1 need presented before Table 2.

4)Figure 3 need presented before Figure 4.

Author Response

Dear editors and reviewers,

Thank you very much for your constructive comments and the chance to improve our manuscript.

We revised our submission to address reviewer comments and the point-to-point responses are listed below:

Reviewer 2

In the manuscript, authors used a deep learning algorithm on patients undergoing weight changes to perform the steatosis quantification. It is an interesting and significative study. Some individual comments were provided below:

Major

  • Why authors collected patients whose weight changes greater than or equal to 5%? please provide some explanation. Similarly, please provide the explanation about the definition of “poor quality” in the sentence “We excluded images with poor quality, without body weight records, or those recorded as dual images.”

Thanks for the important comments. We have added additional explanation to indicate 5 % weight change can decrease liver steatosis significantly and a reference is added. We excluded poor quality images by direct observation (D Tai and T Hsu who have more than 20 years experiences in US studies.) We have added more explanation on this exclusion process as well.

  • Limited sample size is the major limitation of the study. Authors need to provide some explanation in the section of discussion.

Thanks for the notice. We now highlight it in the limitation section of discussion.

Minor

  1. Delete the words “(word count 200)” in the end of the abstract.

Removed.

  1. In the methods, authors need to provide more information about the ethics including if the informed consents were obtained from all the participants.                       The informed consent was waived by IRB because this is a retrospective study.
  2. Table 1 need presented before Table 2.

Fixed, thank you.

  1. Figure 3 need presented before Figure 4. We We rearrange text, thank you.